# A Review of Functional Analysis of Endothelial Cells in Flow Chambers

**DOI:** 10.3390/jfb13030092

**Published:** 2022-07-12

**Authors:** Makoto Ohta, Naoya Sakamoto, Kenichi Funamoto, Zi Wang, Yukiko Kojima, Hitomi Anzai

**Affiliations:** 1Institute of Fluid Science, Tohoku University, 2-1-1 Katahira, Aoba-ku, Sendai 980-8577, Japan; zi.wang.p3@dc.tohoku.ac.jp (Z.W.); kojima.yukiko.r8@dc.tohoku.ac.jp (Y.K.); hitomi.anzai.b5@tohoku.ac.jp (H.A.); 2Graduate School of Systems Design, Tokyo Metropolitan University, 1-1 Minami-Osawa, Tokyo 192-0397, Japan; 3Graduate School of Biomedical Engineering, Tohoku University, 6-6-12, Aramaki Aza Aoba Aoba-ku, Sendai 980-8579, Japan; 4Graduate School of Engineering, Tohoku University, 6-6, Aramaki Aza Aoba Aoba-ku, Sendai 980-8579, Japan

**Keywords:** flow chamber, endothelial cells, coculture techniques, microfluidics, lab-on-a-chip

## Abstract

The vascular endothelial cells constitute the innermost layer. The cells are exposed to mechanical stress by the flow, causing them to express their functions. To elucidate the functions, methods involving seeding endothelial cells as a layer in a chamber were studied. The chambers are known as parallel plate, T-chamber, step, cone plate, and stretch. The stimulated functions or signals from endothelial cells by flows are extensively connected to other outer layers of arteries or organs. The coculture layer was developed in a chamber to investigate the interaction between smooth muscle cells in the middle layer of the blood vessel wall in vascular physiology and pathology. Additionally, the microfabrication technology used to create a chamber for a microfluidic device involves both mechanical and chemical stimulation of cells to show their dynamics in in vivo microenvironments. The purpose of this study is to summarize the blood flow (flow inducing) for the functions connecting to endothelial cells and blood vessels, and to find directions for future chamber and device developments for further understanding and application of vascular functions. The relationship between chamber design flow, cell layers, and microfluidics was studied.

## 1. Introduction

Vascular tissue is composed of many layers; the innermost of which is called the endothelium, which is made of vascular endothelial cells (ECs). Therefore, ECs are constantly contacting the blood flow and are subjected to the flow’s mechanical stress to express their functions. To elucidate this phenomenon, the morphology of vascular ECs was analyzed in vivo [1]. Later, a method for seeding ECs in a chamber was proposed to observe the various functions of ECs. Flow load was applied using the culture medium to simulate blood flow. This method has become indispensable for studying EC functions induced by flow. Recent advances in chamber design technology have led to the development of chambers with ECs and smooth muscle cells (SMCs) cocultured. The relationships between ECs and the function of SMCs and other deeper layers in blood vessels using the chambers have become clear. There is a cascading effect of the influence of blood flow through ECs into the deeper tissues of blood vessels. The chamber-based method has contributed to the elucidation of this cascade. In addition, the development of microfluidic devices through the advancement of microfabrication technology has enabled us to construct a three-dimensional microvascular network that reproduces the microenvironment in vivo and clarifies the dynamics of cell groups (cells).

As described above, the functional expression and application of ECs induced by flow are closely related to the development of many chambers and devices. The purpose of this study is to show the methods of inducing flow for functionalizing ECs by inducing flow and affecting blood vessels. Additionally, this study will find directions for future designs of chambers and device developments for further elucidation and application of vascular functions.

## 2. Materials and Methods

A keyword-based search of Pubmed (https://pubmed.ncbi.nlm.nih.gov) was performed. The search was performed between 23 June and 20 August 2021. The keywords used are shown in Figure 1, and the number of hits for each keyword is shown. The keywords were divided into the three primary categories as shown in Figure 1. The first category is called monolayer of EC. In the chamber, EC are seeded with monolayer cells, and the functions are induced by flow to examine the stimulation in ECs or with ECs. By using assembly techniques, the chambers are constructed as parallel plate, T-chamber, step, cone plate, and stretch, respectively dependent on the induced flow.

The second category is called ‘coculture’, meaning that ECs induced by flows function by flow with SMCs. The third category is called ‘microfluidic’, meaning that the flow channels are fabricated more complexly and more precisely with patterns to use rare, diseased, or patient-specific cells with ECs stimulated by flows.

## 3. Results

### 3.1. Monolayer Chambers

#### 3.1.1. Parallel Chamber

Parallel chambers introduce a unidirectional flow, and the cross-sectional area of the fluid domain is constant, as shown in Figure 2. A layer of EC is created at the bottom of the fluid domain in the chambers. Polystyrene [2], PDMS [3], ULTEM [4], glass for seeding cells in polycarbonate [5], gelatin-coated polyester sheet [6], and fibronectin-coat [7] are often used as the material of the chamber. A pulsating or steady flow was introduced into the chamber. The flow estimations in the chamber were accompanied mostly by computational fluid dynamics (CFD). On the other hands, LDV to experimentally measure the flow was performed and compared with some calculated values by Avari et al. [4], and microparticle image velocimetry was used by Lafaurie-Janvore et al. [8]. In the pulsatile flow, the actin microfilament and NO production of ECs were compared in wall shear stress (WSS) during exercise in vivo [9].

According to the purpose of these studies, we can categorize the EC’s reactions or the activities of signals, including adhesion on the EC. Parallel chambers have been used to study not only the behavior of ECs themselves, but also the relationship between the adsorption of proteins on ECs and activation behavior.

ECs orient and elongate their shapes along with the flow. This has been quantitatively measured by Levesque et al. using the shape index [10]. The results showed that the shape index of bovine ECs almost converged after 24 h at 8.5 Pa WSS. Prasad et al. investigated the response of inositol phosphate levels [11]. Munn et al. investigated cell fluxes under several flow conditions such as steady or saw-tooth patterns [12]. Lafaureie-Janvore et al. showed the polarization of a cell on micron-size lines [8]. Anzai et al. proposed the disturbance effect by the placement of a stent strut on the cell culture in a parallel flow chamber [13]. Wang et al. investigated the cell density in the gap of two stent struts using a parallel flow chamber [14].

The relationship between EC walls, such as endothelium and fluid domain, were studied from signal activation to adhesion on the wall.

Cadroy et al. (1997) studied thrombogenicity using patient blood [15]. Gopalan et al. (1997) studied the emigration of neutrophil at 0.2 Pa [16], and Gosgnach et al. (2000) found that angiotensin converting enzyme is expressed by WSS [17]. Han et al. reported that shear stress induces mitochondrial reactive nitrogen species formation and inhibits the electron flux of the electron transport chain at multiple sites [18]. Popa et al. showed the formation of CD154-induced ultra-large von Willebrand factor (ULVWF) on ECs under 0.25–1 Pa WSS and transmigration of monocytes [19].

Lawrence et al. investigated the adhesion of polymorphnuclear leukocytes (PMNL) to endothelium (HUVEC) under 0.098 Pa [20]. Barabino et al. (1997) studied the adsorption of sickle red blood cells using parallel chambers. Viegas et al. (2011) investigated the adsorption of Methicillin-resistant Staphylococcus aureus on HUVECs in a 0–1.2 Pa parallel plate flow chamber [5]. Ozdemir et al. (2012) investigated the adsorption of soluble fibrinogen-mediated melanoma-polymorphonuclear neutrophils (PMNs) using a commercially available parallel plate flow chamber [21]. Kona et al. (2012) measured the poly (D, L-Lactic-co-glycolic acid) (PLGA) biodegradable nanoparticles uptake into ECs, which are considered to be injured arterial walls under fluid shear stress [22]. Rychak et al. investigated microbubbles adhesion to endothelium with deformation by WSS [23].

Previous studies have also revealed that WSS stimulates mechanosensitive molecules and intracellular signaling pathways, which can induce morphological and functional changes in ECs. For example, Takahashi and Berk showed that 1. 2 Pa WSS caused rapid activation of extracellular signal-regulated kinase (ERK1/2), which is one of the mitogen-activated protein (MAP) kinases and important for changes in EC gene expression [24]. Li et al. reported that phosphorylation of focal adhesion kinase (FAK), which plays a pivotal role in mechano-chemical transduction signaling pathways in ECs, is crucial in shear stress-induced activation of ERK1/2 [25]. Shear stress increases the influx of Ca^2+^, known as a second messenger for the control of a variety of cell functions, in ECs mediated by Ca^2+^ ion channels [26]. Chachisvilis et al. demonstrated that conformational changes in G-protein-coupled receptors (GPCR) and the activation of G-protein, molecular switches changing the activity of GTP, are also induced by shear stress [27].

As summary, parallel plate chambers are widely used to find the cascade and mechanism of EC signals under dynamic response such as adhesion to ECs under uniaxial flow. As for chamber performance, WSS is used between 0.098 Pa and 8.5 Pa. In the future, a further development direction would be to conduct experiments under high WSS, where, for example, cardiovascular diseases are thought to develop, and next-generation chamber design using CFD methods is needed [3,28].

#### 3.1.2. T-Chamber

The T-chamber is a chamber with a T-shaped cross section, as shown in Figure 3. The flow forms an impinging flow at the stagnation point in the center of the T-junction, and a high WSSG is loaded on the EC. This chamber assumes a bifurcation of an artery, which is the frequent site for a cerebral aneurysm. The T-chamber of Meng et al. (2008) has a WSS of 4 Pa and a WSSG of 30 Pa/cm. Sakamoto et al. (2010) investigated the orientation of ECs in chambers with a WSS of 2–10 Pa and a WSSG of 0–340 Pa/cm for 24–72 h under flow loading.

Additionally, Dunn’s group revealed lymphatic ECs are sensitive to WSSG [29,30]. Several WSSGs are produced using a six-well impinging flow chamber and the migration direction of ECs induced by WSSG is changed by the densities of ECs. 

#### 3.1.3. Step Flow Chamber

A step flow chamber is a chamber that has a vertical step expansion at the entrance (Figure 4). This chamber could be used to generate disturbed flow including flow separation and reattachment following vortex. The gradient WSS is generated to mimic the flow condition of a vascular branch. A step flow chamber could study the stimuli of disturbed WSS on EC. Chiu et al. investigated the disturbed flow sustained by activated EC sterol regulatory element binding protein 1 (SREBP1) [31]. The integrins in the SREBP1 mechano-activation play an important role in the modulation of EC lipid metabolism. Bao et al. found a temporal gradient in the shear effect of the NO expression process in EC [32]. Continually using a step flow chamber, Bao et al. also reported that the temporal gradient in shear induces ERK1/ERK2, c-fos, and Cx43 in the EC signaling pathway [33], and temporal gradient in shear effect EC proliferation [34]. Studies found the lipid bilayer in the EC membrane could sense the WSS and change membrane fluidity [35,36,37].

By using an EC-SMC coculture model together with a step flow chamber, Chen et al. found the disturbed flow could induce EC and SMC expressions of adhesion molecules and chemokines, which contributed to the increased white blood cell adhesion and transmigration [38].

#### 3.1.4. Cone Plate Flow Chamber

One form of flow chamber that provides hydrodynamic loading to the surface of cultured cells is the cone-plate type flow chamber, as shown in Figure 5. As a result of Pubmed survey with the keyword terms ‘endothelial cell’ and ‘cone plate’ and ‘wall shear stress’, 20 articles were found. Of these, 17 were flow exposure experiments using ECs.

The cone plate flow chamber system was started in 1981 by Dewey, C.F. et al. [39]. It was developed based on a commercially available cone plate viscometer. In the cone plate type flow chamber, the cone is rotated to generate Couette flow in the small gap between the cone and the plate. Fluid shear stress occurs in a moving viscous fluid as a tractive force against a solid body. Newton’s law of viscosity states that the shear stress τ is proportional to the strain rate: $\tau =\mu \frac{du}{dy}$, where *u* is the fluid velocity and *y* is the distance from the wall. On the rotational device, the equation can be approximated using angular velocity ω and cone angle α as $\tau =\mu \frac{\omega }{\alpha }$, under the assumption that the cone angle is sufficiently small [39]. While blood is a suspension and exhibits non-Newtonian properties, the culture medium is considered as a Newtonian fluid and is assumed to follow the above equations.

Then, the cell layer on the bottom plate was assumed to be exposed by uniform shear. For EC exposure, wall shear up to 120 dyne/cm^2^ has been applied. By fixing the rotation speed and direction of the cone, a constant laminar flow is given [40,41,42,43,44,45,46].

One advantage of the cone plate type flow chamber is that transient flow can be applied using the same device as constant flow. Bongrazio et al. performed turbulent flow exposure (average WSS of 0.6 Pa (6 dyn/cm^2^)) [43]. Franzoni et al. also performed transient flow exposure based on the sine waveform which ranges from 0.5 to 2.5 Pa (from 5 to 25 dyn/cm^2^) of WSS [47]. O’Keeffe et al. performed transient flow exposure based on the waveform obtained by CFD on the right coronary artery [48]. Maroski et al. and Parker et al. also applied a transient waveform [44,49] based on arterial flow profiles: a so-called ‘atheroprone’ profile ranges from −8.9 to 3.7 dyn/cm^2^ (−0.15 dyn/cm^2^ mean) and ‘atheroprotective’ profile with the range from 13.3 to 43.7 dyn/cm^2^ (20 dyn/cm^2^ mean) introduced by Dai, G. et al. [50]. The same as these two articles, Franzoni et al. [51] applied waveform-WSS which involved the inflammatory phenotype of ECs provided by Feaver et al. [52].

Note that the nonlinearity of flow appears because of the secondary flow at the edge of the cone and can alter the shear stress on the plate surface, especially at high shear rate conditions with a larger cone angle. A numerical experiment by Shankaran et al. showed the increase of WSS magnitude with radial position away from the center according to the increase of Re number and cone angle [53]. In their experiment, 0.5° cone did not deviate from the primary flow value. However, 2° cones with a shear rate of 1000 s^−1^ caused the WSS to be ~5.1-fold larger than the primary flow value.

#### 3.1.5. Stretch Chambers

ECs constantly undergo cyclic stretching due to blood pressure applied to the vessel wall. The purpose of using the stretch chamber shown in Figure 6 is to examine ECs loaded by the mechanical role of the wall in addition to being induced by flow. In 1990, Vedernikov et al. first demonstrated that female pigs’ (8–10 weeks of age) left circumflex coronary arteries were directly stretched with a strain gage to examine the tension as a contractile [54]. They examined the effect of the existence of the ECs under stretch on the tension using the ring of the coronary arteries. However, in 1997, Rosaries et al. studied the response of EC collected from calf thoracic aorta by seeding them in a stretch chamber [55]. In this method, 4–25% stretching at 0.5 Hz or 1 Hz is applied to examine EC deformation [56,57], NO generation [58], and Ca^2+^ concentration. In 2008, Katanosaka et al. observed tyrosine phosphorylation and actin dynamics using fibronectin dots in a chamber with a cyclic and uniaxial strain [59]. Hashimoto-Komatsu et al. used the Boyden chamber method to investigate the function of Angiotensin II and its relationship to the function of microtubules in ECs involved in cell-cell adhesion [60].

### 3.2. Summary of Monolayer Chambers

Monolayer chambers can reveal the flow influence on the ECs with other substances. Using a parallel chamber, WSS induced by one-direction flow was applied and the WSS stimulated ECs are the cascade and mechanism of EC signals. T-chambers have WSSG, step flow chambers give vortex, cone chambers give different directions (time dependent on flow), and stretch chambers show the vessel wall deformation. These chambers relate to the representative of arterial geometry such as bifurcation using T-chamber, which is the frequent site for a cerebral aneurysm. The ECs in the chamber have a relation to the diseases. However, the ECs in the monolayer chamber do not have any response to/from/with other cells such as SMC or other organs. The monolayer chamber is missing the physiology of the wall including SMC and the extracellular matrix.

### 3.3. Application of WSS to EC-SMC Coculture

#### 3.3.1. Interaction between ECs and SMCs

As described in the previous chapter, in vitro experiments with various types of flow chambers have revealed the effect of WSS on the morphology and function of vascular ECs. Diverse functions of ECs are responsive to WSS stimuli, such as vasoconstriction/dilation (vascular tone), wall permeability, leukocyte adhesion, thrombogenesis, and vascular wall remodeling. Endothelial responses to WSS are now widely recognized to play a critical role in vascular physiology and pathology such as arteriosclerosis and aneurysms.

In addition to the ECs lining the lumen of arterial walls, SMCs in the middle layer of blood vessel walls, ‘tunica media’, are also important in vascular physiology and pathology. In normal healthy arterial walls, SMCs reside in a quiescent and contractile state, referred to as the ‘contractile type’, and play a major role in vascular tone. However, in the medial degeneration site widely recognized as a pathologic feature of aortic diseases such as arteriosclerosis and aneurysms, SMCs dedifferentiate into a proliferative and synthetic state (synthetic phenotype) and exhibit low contractility and high proliferative and migration ability, with secretions of physiological activity factors and extracellular matrix (ECM) proteins. Since SMCs are not directly exposed to blood flow, the response of ECs to the WSS condition causes changes in the phenotype and function of SMCs through cell–cell interaction (cross-talk) [61]. Investigations focusing on cell–cell interactions between ECs and SMCs under WSS conditions have been performed to understand the role of EC response to WSS in vascular physiology and pathology in more detail.

#### 3.3.2. EC and SMC Coculture Models for WSS Experiments

Coculture assays have been demonstrated to study the cellular interactions between ECs and SMCs in vitro. The coculture systems for investigating the effects of WSS on the EC-SMC interactions have also been developed. In this review, focusing on the interaction between ECs and SMCs in vascular physiology, we searched for the keyword terms ‘coculture’ or ‘cocultured’ in addition to ‘endothelial’, ‘shear stress’, and ‘smooth muscle’ in Pubmed (Figure 1). From the 63 identified articles from 1995 to July 2021, excluding non-English journals and reviews, we examined 45 studies that performed WSS experiments for coculture of ECs and SMCs.

The coculture methods used in the examined studies can be classified as follows (Figure 7).
Double-side type (flat and tubular types; Figure 7A,B): ECs and SMCs are cocultured on the opposite sides of a porous membrane (flat type, 20 cases (44.4%), tubular type, 6 cases (13.3%), total of 26 cases (57.8%));Single-side type (Figure 7C): ECs and SMCs are cocultured on the same surface (3 cases (6.7%));Direct culture type (Figure 7D): ECs are directly cultured above a pre-cultured SMC layer (7 cases (15.6%));3D culture type (Figure 7E): ECs are cultured on type I collagen gel or the other types of ECM gel containing embedded SMCs (8 cases (17.8%));Another type: ECs are cultured on the inside of a culture insert, and SMCs are cocultured on the bottom of the well in which the culture insert was placed (1 case (2.2%)).

The ‘double-side type’ is the most common method for WSS experiments for coculture of ECs and SMCs among the studies examined, and many of these studies used originally-developed parallel plate flow chambers, which can incorporate commercially available cell culture inserts and apply WSS to ECs on the bottom of the culture inserts [62,63,64,65,66,67,68,69,70,71,72]. In some cases of the double-sided type, tubes or capillaries made of porous materials were also used [73,74,75,76,77]. There seems to be no essential difference in the EC-SMC cross-talk and WSS acting on ECs compared to the method using culture inserts, but in the tubular type, ECs are surrounded by SMCs, which is a similar environment to that of blood vessels. The ‘direct culture type’, ‘3D culture type’, and ‘single-side type’ are generally formed in the flat plane shape of cocultures in the culture dish, and the parallel plate flow chambers are commonly adopted in these types of coculture methods. Experiments using a cone (disk) plate flow chamber [69,78] and perfusing culture medium into an original tubular substrate having a honeycomb cross-sectional shape [79] have also been reported as methods for application of WSS to EC-SMC cocultures.

#### 3.3.3. Responses of ECs and SMCs to WSS under Coculture Conditions

Many studies used EC-SMC coculture experiments to understand the mechanism of arteriosclerosis. Therefore, these coculture studies also investigated the effects of WSS environments lower than the physiological levels of arteries, and have shown that WSS applied to ECs causes functional changes in SMCs not exposed to WSS. In addition, it has also been reported that the effects of WSS on ECs cocultured with SMCs are different from those on monocultured ECs, and factors that act as signal transmitters in the intercellular cross-talk in coculture environments. These studies have then revealed that the conditions of static culture and WSS lower than ~0.5 Pa have atheroprone effects on the behavior and functions of ECs and SMCs related to the formation and development of arteriosclerosis, and physiological levels of WSS induce atheroprotective responses in cells. Increased migration and proliferation of SMCs have been observed in arteriosclerotic lesions, and it has been reported that the application of 1 Pa of WSS on ECs suppresses the growth of SMCs not exposed to WSS compared to static conditions [66,69]. The WSS applied to ECs also suppresses the migration of SMCs [75,80,81], and it was revealed that nitric oxide (NO) produced by ECs in response to WSS plays a critical role in the suppression of SMC migration [80]. An increase in leukocyte adhesion and invasion of the blood vessel wall has also been observed in arteriosclerosis pathology, and the effects of coculture and WSS environment have been reported on this phenomenon. Coculturing with SMCs increases the expression of adhesion proteins such as intercellular adhesion molecule-1 (ICAM-1), vascular cell adhesion molecule-1 (VCAM-1), and E-selection on the surface of ECs, as well as the expression and secretion of chemokines and cytokines such as monocyte chemotactic protein-1 (MCP-1), growth-related oncogene-α (GRO-α), and interleukin-8 (IL-8) that promote leukocyte migration, and these expressions and secretions induced by the coculture are suppressed by a physiological level of 1.5 Pa WSS condition [62,63,69,82]. It has also been shown that WSS applied to ECs suppressed leukocyte invasion stimulated by an SMC coculture environment [83]. The WSS conditions of ECs also affect the phenotype of cocultured SMCs. The expression of contractile markers, smooth muscle α-actin (SMα-actin), SM-myosin heavy chain (SM-MHC), and calponin in SMCs was increased by 1.2 Pa of WSS applied to cocultured ECs [69]. Hastings et al. exposed ECs to atheroprone oscillatory WSS and pulsatile WSS, simulating physiological conditions, and evaluated the phenotypes of ECs and SMCs. As a result, oscillatory WSS decreased the expression level of the physiological state of quiescent phenotypic ECs’ markers such as endothelial NO synthase (eNOS) and Tie2, and similarly decreased the contractile markers, SMα-actin, SM-MHC, and myocardin, of SMCs compared to the physiological conditions [63]. In cross-talk between ECs and SMCs under WSS conditions, the roles of prostacyclin (PGI_2_) [69] secreted by ECs, platelet-derived growth factor (PDGF)-BB, and transforming growth factor-β1 (TGF-β1) [67] have also been shown.

#### 3.3.4. Advanced Applications of EC and SMC Coculture Models

As stated above, coculture studies have revealed that EC-SMC interactions have crucial roles in vascular physiology and pathology under WSS conditions, especially in the pathogenesis of arteriosclerosis. Extending the work to include constructing tissue-engineered blood vessels and developing screening assay platforms for drug discovery as an alternative to animal testing, the improvement of EC-SMC coculture systems and the conduct of flow exposure experiments lasting longer than the typical conventional 24 h have also been performed. Cultured SMCs generally show the synthetic type while the normal state of SMCs in the arteries shows a contractile phenotype, and it has been pointed out that the effects of synthetic SMCs on cocultured ECs are different from those of contractile cells [84,85]. In addition to the phenotype of cocultured SMCs, they have focused on the effect of mechanical properties of culture inserts widely used in ‘double-side type’ coculture and the suppression of direct contacts between ECs and SMCs by the inserts [84,85]. They have conducted the ‘direct culture type’ coculture of ECs and SMCs. Some studies have demonstrated WSS experiments with EC-SMC coculture, which was constructed with SMCs differentiated into a contractile type by culturing with a serum-free culture medium in advance [86,87]. Longer-term WSS experiments using bioreactors have also been performed. For the purpose of investigating molecular mechanisms related to angiogenesis, vascular wall remodeling, and vascular disease, Janke et al. developed a bioreactor as an ‘artificial artery’, in which ECs were cultured on the inside of porous capillaries and SMCs on the outside, and conducted the application of WSS for 5 days [77]. To evaluate blood-brain barrier (BBB) characteristics in the cerebrovascular network, Cucullo et al. made an artificial vascular system mimicking cerebral capillary and venous segments by connecting a vein model in which ECs and SMCs were cocultured and a capillary model composed of endothelial and astrocyte coculture [76]. They applied WSS of 0.3 Pa to the vein model as well as WSS of 1.6 Pa to the capillary model for 3 weeks, and examined the relationship between the formation of cell–cell adhesion as assessed by transendothelial electronic resistance (TEER) and endothelial permeability in the cerebrovascular network and the experimental period. Since in vitro cell culture experiments have strong advantages in studying cell–cell cross-talk at the molecular level, coculture WSS experimental systems not only with ECs with SMCs but also with other types of cells such as pericytes, astrocytes, and valve interstitial cells will be more important for a detailed understanding of physiology and pathology, as well as for regenerative medicine and drug discovery.

Recently, it has been pointed out that high WSS has an effect on pathology such as aneurysms and arterial dissection [88,89], but physiological levels of up to 4 Pa WSS have been examined in coculture studies of ECs and SMCs because these WSS experiments have also been performed mainly for atherogenesis, and one study showed the effects of WSS up to about 10 Pa on EC-SMC coculture [90]. Since cross-talk between ECs and SMCs is considered to play an important role in pathogenesis associated with higher WSS conditions as well, high WSS experiments for coculture will also be required for elucidating these pathologies.

### 3.4. Cellular Experiments of ECs with Microfluidic Devices

#### 3.4.1. Microfluidic Cellular Experiments

Due to the advancement of microfabrication technology, cellular experiments using microfluidic devices, which culture cells inside microchannels as shown in Figure 7, have been performed to observe cellular dynamics since the 2000s. We searched for keyword terms ‘microfluidic’ or ‘microfluidic device’ with ‘endothelial cell’ and ‘shear stress’ in Pubmed, resulting in the identification of 278 or 94 articles, respectively (Figure 1). In addition, in order to include state-of-the-art technology of microfluidic cellular experiments into this review, search for keyword terms ‘microfluidic device’ in addition to ‘vascular network’ identified 129 articles. Here, referring to the search results as appropriate, we provide an overview of cellular experiments using microfluidic devices.

In fabricating microfluidic devices, a convex channel pattern is first created on a silicon wafer using SU-8 photolithography or on a plastic polymer such as ABS resin by milling machine. The channel pattern is then transferred to polydimethylsiloxane (PDMS) [91] or a hydrogel [92,93] by soft lithography. Inlets and outlets are punched in the PDMS or hydrogel mold to access each channel, and a layer of the same material or a cover glass is bonded to the channel-patterned surface to form the channels. The devices consist of channels for cell culture, for hydrogels to mimic an ECM, and for loading various conditions on the cells (media and gas channels). Hereby, the control of environmental factors (mechanical and chemical stimuli) on cells is achieved, and cell dynamics under conditions that reproduce in vivo microenvironments can be evaluated. The usage of a microfluidic device for cellular experiments can save rare cells, such as patient-derived cells and stem cells, and expensive experimental reagents. Cell adhesion area and the fluid volume in a microfluidic device are smaller than those in cell culture dishes and wells, and small amounts of cells and reagents are sufficient for performing experiments. The coculture of multiple types of cells and their three-dimensional culture are feasible in the device. In addition, since the device is fabricated using transparent materials, it enables high-resolution and real-time observation of cell dynamics.

#### 3.4.2. Microfluidic Devices for Shear Stress Applications

Many experiments have been performed with ECs in confluency in a microfluidic device to investigate their dynamics under flow exposure. Various shear stresses were applied to the ECs cultured in the channel by flowing the cell culture medium using pumps such as syringe pumps [94], roller pumps [95], or by applying hydraulic head pressure between the inlet and outlet of the channel [96]. To generate various levels of steady shear stress at the same time, other than directly controlling the shear stress by adjusting the flow rate of the cell culture medium, there are ways to change the geometry of the flow channels in the device [97,98]. The width or height of a single channel can be continuously [30,99,100,101,102,103] or stepwisely [104,105,106,107] enlarged or reduced. Otherwise, the channel size is changed by branching a channel into multiple channels [108,109,110,111] or by adjusting the circular diameter [112]. Even if the flow rate of the cell culture medium flowing in the entire channel is constant, the flow velocity varies with the cross-sectional area, and a large shear stress can be generated at a location with a narrow cross-sectional area. The shear stress can also be controlled by controlling the flow rate of the cell culture medium with the setting flow resistance of branching channels [113,114,115,116]. The effect of spatiotemporal gradients of shear stress on cells can also be studied by varying the shear stress within the flow channel [117]. In order to generate unsteady shear stress, it is common to combine a microfluidic device with the supply of cell culture medium by a pump [118,119]. Experiments have been conducted to load shear stress on the ECs by generating periodic flow, such as beating [120,121]. Pumpless microfluidic devices have also been proposed to periodically generate bidirectional flow [122]. These microfluidic methods enable us to simulate both physiologically healthy situations of shear stress and situations at the site of onset of arteriosclerosis and other diseases.

In experiments using microfluidic devices, it is possible to observe the dynamics of ECs under multiple stimuli in addition to flow exposure. By fabricating the device with elastic materials, the device itself can be deformed by stretching and shrinking to exert mechanical stimuli over the ECs [106]. Changes in the components such as adding glucose [100], vascular endothelial growth factor (VEGF) [123], tumor necrosis factor (TNF-α) [124,125], adenosine triphosphate (ATP) [117], or EDTA [126] in the cell culture medium yield to exert chemical stimuli over the cells. Experiments combined with micropatterning can also be conducted by modifying the surface properties of the substrate like hardness and hydrophilicity by coating with hydrogel or other materials [127,128] or by plasma treatment [8]. Furthermore, the effects of oxygen tension on ECs can be studied by manipulating the dissolved gas components in the cell culture medium [125,129,130]. As for a device structure, it has been proposed that a channel for culturing ECs and another separated channel be located sandwiching a membrane with nano-sized pores [131,132,133] or a hydrogel. This structure allows a coculture with different types of cells, such as pericytes or astrocytes that support blood vessels [134], cancer cells that promote angiogenesis [135], and epithelial cells that exist on the other side of the monolayer of ECs [136]. Consequently, it is possible to observe the dynamics of ECs under the interaction with other types of cells. Furthermore, observation of effects of blood cells (platelets, leukocytes, and red blood cells) [112,137] or parasitic protozoa and bacteria such as toxoplasma [138] are injected into the channel where ECs are cultured, considering it as a blood vessel.

#### 3.4.3. Experiments with EC monolayer

With monolayers of ECs formed in the microfluidic device, morphological changes of the cells in response to environmental factors, including shear stress, have been observed. It has been shown that cells orientate in the flow direction by flow exposure, but orientate in a direction orthogonal to the flow direction when the shear stress is very high. The presence or absence of flow affects the differentiation and phenotype of ECs and alters their properties. The expression of intercellular adhesion molecules such as ICAM-1 and cell-substrate adhesion molecules, as well as cytoskeletal changes by actin filament, play important roles in the morphological changes of cells, and they vary according to the magnitude and period of the flow exposure. Although ECs forming a monolayer are distributed like paving stones, individual cells do not lose their motility, and random collective migration is observed [8,95,130]. By acquiring time-series microscopic images of this collective migration and analyzing them by particle image velocimetry, the migration velocity of the cells and the strain (traction force) generated in the monolayer can be obtained to clarify the dynamic characteristics [105]. The wound healing assay, in which a monolayer of cells cultured on a dish is scratched and then the cells recovering the damaged area are observed, has frequently been used to investigate collective migration. In an experiment using a microfluidic device, a similar wound healing assay that chemically damages a monolayer of ECs by flowing trypsin solution under fluid control is proposed [139]. Moreover, collective migration of ECs is a necessary process in sprouting at the beginning of angiogenesis. By placing a hydrogel that mimics an ECM in a microfluidic device and forming a monolayer of ECs at the interface, the effects of interstitial flow in the ECM and shear stress on angiogenesis can be evaluated [140]. Furthermore, it is possible to quantitatively evaluate the permeability of a monolayer of ECs. The permeability can be measured by quantifying the diffusion of fluorescence-labeled dextran, which is added to the cell culture medium, goes through the monolayer formed on a hydrogel, and diffuses in the gel [114,133,138,141]. Alternatively, the same as the TEER method using a transwell, the electrical resistance between the upper and lower channels separated by a monolayer of ECs formed on a membrane with nano-sized pores can provide permeability [118,132,135].

#### 3.4.4. Experiments with Microvascular Networks and Their Perspective

Research using microfluidic devices has also established a method to construct a three-dimensional microvascular network like capillaries [142]. By culturing ECs densely mixed in a hydrogel such as fibrin gel, vasculogenesis occurs and a microvascular network is formed [143]. To stabilize the microvascular network, cells such as pericytes and fibroblasts should be mixed with ECs at an appropriate density. Additionally, it has been proposed to mix astrocytes found in the central nervous system to reproduce the BBB of blood vessels in the human brain [144]. The permeability of the microvascular network formed in microfluidic devices has been evaluated, as well as the changes in response to cell composition, chemical stimuli, and shear stress. Additionally, microvascular networks are utilized to observe the intravascular invasion and extravasation of cancer cells in a cancer microenvironment [145,146]. Furthermore, nanoparticles are injected into microvascular networks for drug delivery purposes [147].

Cell experiments using microfluidic devices have evolved into organ-on-chips that reproduce the functions of organs and in vivo tissues by integrating various channel structures and cells. One notable example is lung-on-a-chip, which mimicked a microenvironment in an alveolus by coculture of endothelial and epithelial monolayers with sandwiching a porous membrane set in a microchannel [148]. The chip yielded observation of various cell dynamics in the presence of blood and air flows and stretching. Additionally, devices have been proposed to mimic the function of the entire human body by connecting the functions of multiple organs. The use of such microfluidic devices makes it possible to perform cellular experiments under conditions that reproduce the microenvironment in vivo, and it is expected to be utilized for drug screening for various diseases. The importance of microfluidic devices for cellular experiments will increase in the future as a research tool that contributes to the 3Rs (Replacement, Reduction, and Refinement) in animal experiments for medical research, including drug discovery.

## 4. Conclusions

This review revealed that the flow-inducing functions are connected to EC_S_ and SMCs in blood vessels as shown in Table 1. The studies using flow chambers are elucidating from the response of ECs themselves on surfaces in direct contact with flow to the response of cells in the wall to signals from ECs. These chamber studies have shown that the effects of flow are transmitted in a cascade from ECs to cells in the wall. The relationship with organs other than arteries is also spotted using the flow chambers. 

## Figures and Tables

**Figure 1 jfb-13-00092-f001:**
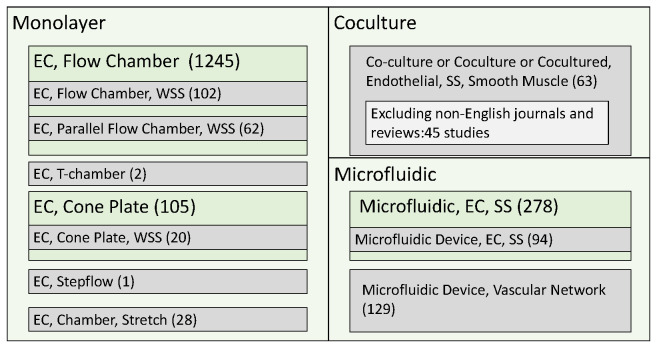
The main categories of paper searches using keywords in Pubmed.

**Figure 2 jfb-13-00092-f002:**
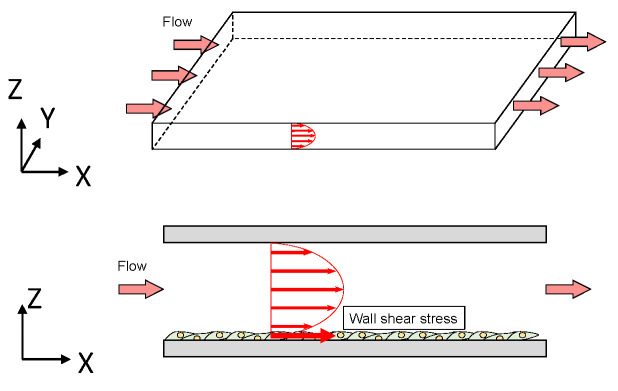
Flow domain in a parallel chamber. The arrows in the flow domain show the velocity distribution.

**Figure 3 jfb-13-00092-f003:**
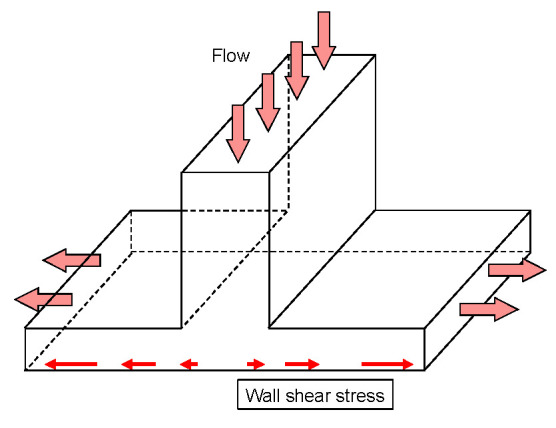
Flow domain in the T-chamber.

**Figure 4 jfb-13-00092-f004:**
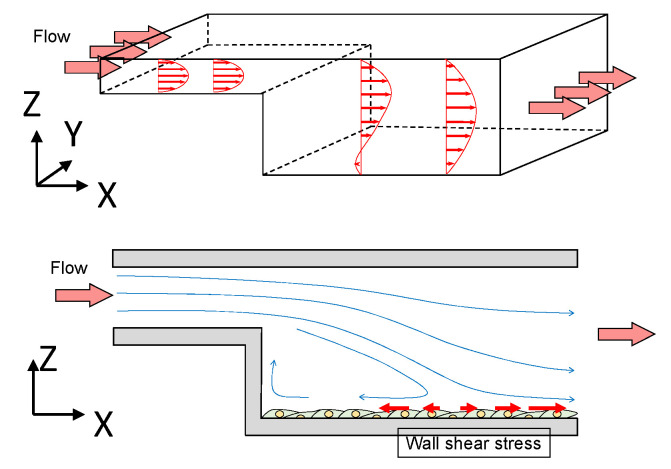
Flow domain in a step flow chamber. The red arrows in the flow domain show the velocity distribution, and the blue arrows show the flow patterns in the flow domain.

**Figure 5 jfb-13-00092-f005:**
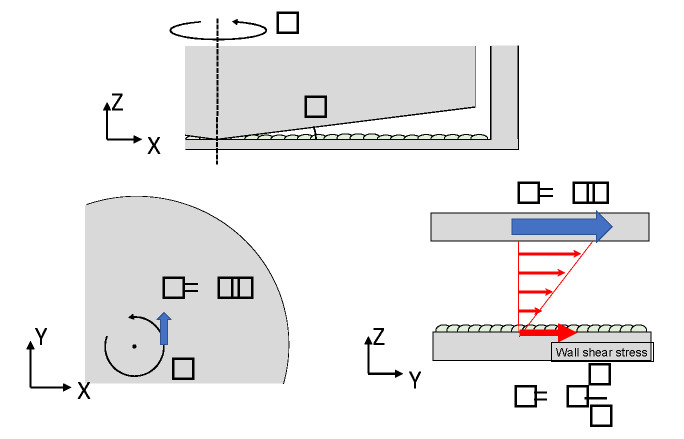
Flow domain in a cone plate flow chamber. The red arrows in the flow domain show the velocity distribution, and the blue arrows show the direction of the cone.

**Figure 6 jfb-13-00092-f006:**
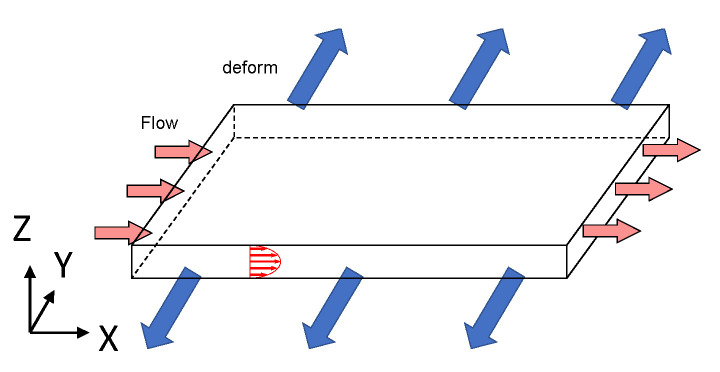
Flow domain in a stretch chamber. The red arrows show the velocity distribution in the flow domain, and the blue arrows show the direction of the wall of chamber.

**Figure 7 jfb-13-00092-f007:**
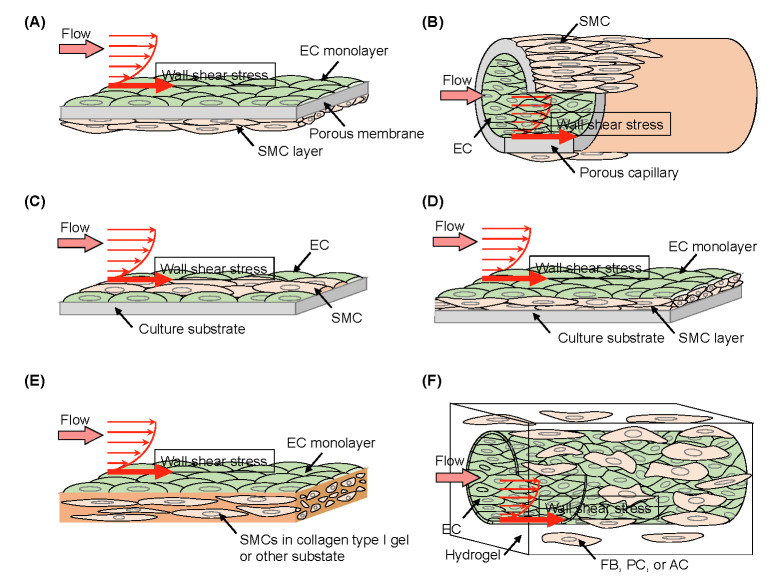
Schematic illustration of types of EC and SMC coculture models for WSS experiments found in the literature. (**A**) Coculture of ECs and SMCs on the opposite sides of a flat porous membrane. (**B**) Coculture of ECs and SMCs on the opposite sides of a porous tube. (**C**) Coculture of ECs and SMCs on the same surface (mixed or arranged). (**D**) Coculture of ECs directly above pre-cultured SMCs. (**E**) Coculture of ECs on the surface of type I collagen gel or the other types of ECM gel containing SMCs. (**F**) Coculture of ECs forming a capillary-like structure in a hydrogel with surrounding fibroblasts, pericytes, and/or astrocytes. In either of the models, except for C, WSS exerted only on ECs. FB, fibroblast; PC, pericyte; AC, astrocyte.

**Table 1 jfb-13-00092-t001:** Chambers with flow character and EC responses.

	Signal and Response by Flow	Name of Chamber	Flow Character	ECs on …
Monolayer	Inside ECs/With ECs	Parallel (Figure 2)	One direction	Rigid wall
		T-chamber (Figure 3)	WSSG	Rigid wall
		Step (Figure 4)	Vortex	Rigid wall
		Cone plate (Figure 5)	Couette flow	Rigid wall
		Stretch (Figure 6)	One direction	Deformed wall
Coculture with SMC	Cross-talk with SMC	Double-side flat (Figure 7A)	One direction	porous membrane with SMC in the oppposite side
		Double-side 3D (Figure 7B)	3D tubular	
		Single-side (Figure 7C)	One direction	Rigid wall with SMC
		Direct culture (Figure 7D)	One direction	Directly on SMC
		3D culture (Figure 7E)	One direction	Collagen type 1 gel with SMC
Microfluidic cell culture	Cross-talk with other cells via ECM	Another type (Figure 7F)	3D tubular	In hydrogel

## Data Availability

Not applicable.

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
