# Peer review of "A Review of Functional Analysis of Endothelial Cells in Flow Chambers"

_jfb, 2022, doi:10.3390/jfb13030092_

Round 1

Reviewer 1 Report

In their manuscript the author performed an interesting review of the different types of flow chambers use for endothelial cell culture and their applications.

I just have some minor comments/questions for the authors :

  • In the part 3.1.1.1 "The reaction of the EC itself",  the authors talked about the flow-induced modifications of ECs shape and organization. It would be great if the the authors also add some data about the main cellular signaling pathways modified in the EC by the flow (i.e Erk/Akt pathways...)
  • The part 3.1.2 "T-chamber" is a little bit short. For the other type of flow chamber the authors described well how and why they have been designed and which physiological situation they are trying to mimic. Such kind of data are lacking for the T-chamber so the readers cannot easily understand if and in which cases this kind of chamber is used.
  • I was wondering if the authors could check if there was any flow chambers used on lymphatic endothelial cells.  Indeed these cells belong to another type of vessels and are submitted to a very particular flow mediated by the contraction of lymphangions. I thus would like to know if there are studies performed on these cells with flow chambers.

Besides these points I would like to thank the authors for their work and I am positive for this manuscript to be published

Author Response

In the part 3.1.1.1 "The reaction of the EC itself", the authors talked about the flow-induced modifications of ECs shape and organization. It would be great if the the authors also add some data about the main cellular signaling pathways modified in the EC by the flow (i.e Erk/Akt pathways...)
→I am thankful for your suggestion. We added the main cellular signaling pathways in the EC by the flow during L133-144 at page 4 with yellow line. Please check it.

The part 3.1.2 "T-chamber" is a little bit short. For the other type of flow chamber the authors described well how and why they have been designed and which physiological situation they are trying to mimic. Such kind of data are lacking for the T-chamber so the readers cannot easily understand if and in which cases this kind of chamber is used.
→I am thankful for your suggestion. I added with yellow line “This chamber assumes a bifurcation of artery, which is the frequent site for a cerebral aneurysm.” at L 153-154 at page 4 . Please check it.

I was wondering if the authors could check if there was any flow chambers used on lymphatic endothelial cells. Indeed these cells belong to another type of vessels and are submitted to a very particular flow mediated by the contraction of lymphangions. I thus would like to know if there are studies performed on these cells with flow chambers.
Besides these points I would like to thank the authors for their work and I am positive for this manuscript to be published
→I am thankful for your suggestion. We added several sentences for lympatic ECs in the T-chamber section at L 158-161, page 4.

Reviewer 2 Report

Dr Ohta et al present a review of Functional Analysis of Endothelial Cells in Flow 2 Chambers. The work is interesting, comprehensible and well written. I have just missed in the manuscript some information about those devices that allow measure interaction between two organs through two different chambers (f.i ECs in lungs). Please include some additional information about those possibilities

Minnor

-line 139 . Please specify what it is  CFD method.

-Line 173. It sounds strange include in a review unpublished results  

-Around line 257 the cross talk needs to be referenced. F.i an excellent review was published last year about.  Cellular Crosstalk between Endothelial and Smooth Muscle Cells in Vascular Wall Remodeling. Méndez-Barbero N, Gutiérrez-Muñoz C, Blanco-Colio LM.Int J Mol Sci. 2021 Jul 6;22(14):7284. doi: 10.3390/ijms22147284.

Author Response

I have just missed in the manuscript some information about those devices that allow measure interaction between two organs through two different chambers (f.i ECs in lungs). Please include some additional information about those possibilities

I am thankful for your suggestion. I added the lung-on-a-chip. “One notable example is lung-on-a-chip which mimicked a microenvironment in an al-veolus by coculture of endothelial and epithelial monolayers with sandwiching a porous membrane set in a microchannel [142]. The chip yielded observation of various cell dynamics in the presence of blood and air flows and stretching.” P12 L554-557 with yellow line. Please check it.

Minnor

-line 139 . Please specify what it is  CFD method.

->It is already written in L90. Please check it.

-Line 173. It sounds strange include in a review unpublished results  

->I deleted it.

-Around line 257 the cross talk needs to be referenced. F.i an excellent review was published last year about.  Cellular Crosstalk between Endothelial and Smooth Muscle Cells in Vascular Wall Remodeling. Méndez-Barbero N, Gutiérrez-Muñoz C, Blanco-Colio LM.Int J Mol Sci. 2021 Jul 6;22(14):7284. doi: 10.3390/ijms22147284.

->

I am thankful for your suggestions. I added it.

Reviewer 3 Report

The present review summarizes research in the field of endothelial cell. The field has a wide area of applicability such as general physiopathology, pharmacodynamics and pharmacokinetics to very specific cancer metastasis research. The authors divide their findings in three meaningful categories. They review a high number of articles on the subject from older articles to very recent ones. Even if some of them are older, the authors use them to describe the evolution of this field of the research.

It is written in a clear and concise manner and it provides an overview of this field. It contains figures that help the reader to better understand the article and that increase the reader audience.

However, in sections 3.1.1.1, 3.1.1.2, 3.1.1.3, some works are just presented but their findings are not described. The authors could elaborate a bit more the research presented in these sections. 

The first part of the sentence in line 29, “The purpose of this study is to summarize the blood flow (flow inducing) for the functions connecting to endothelial cells and blood vessels” needs to be rephrased in a clearer manner. What are you summarizing? (e.g.: The purpose of this study is to summarize the blood flow functions connected to endothelial cells and blood vessels…)

In the first sentence in the introduction, line 41-42, there is an error: “endometrium”. It should be endothelium.

Materials and Methods is not a section in the article, it should be renamed to schematic of the review, and explain better why the authors choose to divide the article in this manner. Move the first two paragraphs from results (row 73 to 81) to section 2.

The conclusions are missing. Would be necessary some assessments from the papers  used to write the review, e.g., which is the best chamber.

Author Response

However, in sections 3.1.1.1, 3.1.1.2, 3.1.1.3, some works are just presented but their findings are not described. The authors could elaborate a bit more the research presented in these sections.

I am thankful for your suggestion. I added a summary for the parallel chamber at the L145-150, page 4 with the yellow line. Please check it.

The first part of the sentence in line 29, “The purpose of this study is to summarize the blood flow (flow inducing) for the functions connecting to endothelial cells and blood vessels” needs to be rephrased in a clearer manner. What are you summarizing? (e.g.: The purpose of this study is to summarize the blood flow functions connected to endothelial cells and blood vessels...)

The purpose of this study is to show the flow-inducing functions connected to endothelial cells and blood vessels. This review revealed that the flow effects stimulated the function of ECs itself to a vascular network. The studies using flow chambers are elucidating from the response of ECs themselves on surfaces in direct contact with flow to the response of cells in the wall to signals from ECs. I added this summery in the conclusion at L 544-550 page 13 with yellow lines. Please check it.

In the first sentence in the introduction, line 41-42, there is an error: “endometrium”. It should be endothelium.

I am thankful for your suggestion. I modified it.

Materials and Methods is not a section in the article, it should be renamed to schematic of the review, and explain better why the authors choose to divide the article in this manner. Move the first two paragraphs from results (row 73 to 81) to section 2.

I am thankful for your suggestion. I modified it.

The conclusions are missing. Would be necessary some assessments from the papers used to write the review, e.g., which is the best chamber.

I am thankful for your suggestion. I added the conclusion at L 544-550 page 13 with yellow lines.

However, my intension of this review is not to say the best chamber. But I want to say the capability of chamber under the mimicking the function of ECs with other organs.

Therefore, I will not add the best chamber.

Reviewer 4 Report

This paper reviewed functional analysis of endothelial cells in different flow conditions. The message of the Review paper is too general. The Authors should have discussed more in detail and provide their own perspectives, not just find the papers and cite them.

Authors should discuss the search section, “2. Materials and Methods”.  Why the keywords are important? What is the implication of this search?

Authors just mention the papers and make references, for example 3.1.1.2.  (many other sections), I would recommend to discuss more in depth.

I also missed the biological implications such as how the flow affects the cell morphology and function of the vascular ECs, study of cytokine release in different conditions and drug testing.

Also, a comparison table would be appreciated for Readers to get direct information.

Unless the paper is not significantly changed, I would not accept this reviewer paper in the Journal of Functional Materials. 

Author Response

This paper reviewed functional analysis of endothelial cells in different flow conditions. The message of the Review paper is too general. The Authors should have discussed more in detail and provide their own perspectives, not just find the papers and cite them.
Authors should discuss the search section, “2. Materials and Methods”. Why the keywords are important? What is the implication of this search?
Authors just mention the papers and make references, for example 3.1.1.2. (many other sections), I would recommend to discuss more in depth.

I am thankful for your pointing. I added the following summary of monolayer section.

Monolayer chamber can reveal the flow influence on the ECs with other substances. Using parallel chamber, WSS induced by one direction flow was applied and the WSS stimulated ECs cas the cascade and mechanism of EC signals. T-chamber has WSSG, step flow chamber gives vortex, cone chamber gives different directions (time depended flow) and stretch chamber shows the vessel wall deformation. These chambers relate to the representative of arterial geometry such as bifurcation using T-chamber, which is the frequent site for a cerebral aneurysm. And the ECs in the chamber has a relation to the diseases.  However, these ECs in the monolayer chamber don’t have any response to/from/with other cells such as SMC or other organs. The monolayer chamber is missing the physiology of wall including SMC and extracellular matrix.

Please check L 254-264 at page 7.

I also missed the biological implications such as how the flow affects the cell morphology and function of the vascular ECs, study of cytokine release in different conditions and drug testing.

I am thankful for your points. We already put some sentences (L 359, page 9). Please check it.

Also, a comparison table would be appreciated for Readers to get direct information.

I am thankful for your kind suggestions. I added one table to summarize this paper.

Unless the paper is not significantly changed, I would not accept this reviewer paper in the Journal of Functional Materials.

I am sorry for our misunderstanding. I added several references, material and methods, summary of parallel chambers, lymphatic EC, and intended use of each chamber, and conclusions of this review with table. I hope to cover your requests.

Round 2

Reviewer 3 Report

It's ok. The authors corrected the manuscript according to my observations.

Reviewer 4 Report

The authors addressed the comments. I would accept the paper.